# Maternal occupational exposures during early stages of pregnancy and adverse birth outcomes in the NINFEA birth-cohort

Antonio d'Errico [1,2]*, Maja Popovic[1], Costanza Pizzi[1], Giovenale Moirano[1], Chiara Moccia[1], Lorenzo Richiardi [1], Milena Maule [1]*

1 Department of Medical Sciences, Cancer Epidemiology Unit, University of Turin and CPO-Piemonte, Turin, Italy, 2 Institute for Risk Assessment Sciences, Utrecht University, Utrecht, The Netherlands

* antonio.derrico@unito.it (Ad); milena.maule@unito.it (MM)

## Abstract

### Objectives

Maternal occupational exposures during early pregnancy can be detrimental to foetus health and have short- and long-term health effects on the child. This study examined their association with adverse birth outcomes.

### Methods

The study included 3938 nulliparous women from the Italian NINFEA mother-child cohort. Their occupational exposures during the first trimester of pregnancy were assessed through prospectively collected questionnaire information and job-exposure matrices. Associations between maternal exposures and birthweight, preterm birth, and delivery by caesarean section were analysed by multivariable linear and logistic regression models. An exploratory factor analysis was carried out to explore co-exposure profiles in association with birth outcomes.

### Results

Women exposed to passive smoking at work and those who reduced their working hours during pregnancy were found to have an increased likelihood of all analysed birth outcomes. Children of mothers performing a demanding work were less likely to be born preterm [OR 0.72 (95% CI 0.54 to 0.95)] and more likely to have a higher birthweight [β = 40.4 g (95% CI 7.5 to 73.4)]. Maternal exposures to heat and dust were associated with a lower birthweight [β = -160.1 g (95% CI -299.6 to -20.7)] and increased odds of caesarean section [OR 6.99 (95% CI 2.36 to 25.47)], respectively.

### Conclusions

This study provides some evidence of the selection of healthy population into the workforce and of association between work-related passive smoking, heat and dust and adverse birth outcomes.

**Data Availability Statement:** Data cannot be publicly shared due to legal constraints, patient privacy concerns, and third-party ownership by the

University of Turin and Azienda Sanitaria Ospedaliera "San Giovanni Battista" of Turin. However, you can access the introduction and details of the dataset through the NINFEA birth-cohort study website at https://www.progettoninfea.it/. Alternatively, you can request access to the dataset by emailing the director of the NINFEA cohort study at lorenzo.richiardi@unito.it or contacting the institutional email info@progettoninfea.it, or by reaching out directly via the https://www.progettoninfea.it/contact_us webpage. You can also contact the board responsible for the NINFEA cohort data by phone at +39 0116334673, or by mail at Epidemiologia dei Tumori, Via Santena 7 - 10126 Torino, Italy.

**Funding:** LR received funding for the NINFEA birth-cohort study from: European Union's Horizon 2020 research and innovation programme ATHLETE, grant agreement number 874583 (https://athleteproject.eu/); European Union's Horizon 2020 research and innovation programme LIFE-CYCLE project, under grant agreement no. 733206 (https://lifecycle-project.eu/); Compagnia San Paolo Foundation (https://www.compagniadisanpaolo.it/it/). The funders did not play any role in our study.

**Competing interests:** The authors have declared that no competing interests exist.

## Introduction

The developmental origins of health and disease hypothesis [1] and the human exposome concept [2] are frameworks that focus on understanding the complex interactions between environmental exposures and life-course health outcomes. Foetal and early life are time windows of vulnerability to exposures that may influence the short- and long-term risk of disease. Maternal exposures during pregnancy include a large set of factors such as those related to lifestyle and diet, the strictly understood environment (e.g., air pollution and meteorological parameters), and the full range of occupation-related exposures, including chemical, biological, and physical ones, as well as those related to psychosocial and ergonomic factors.

In many European countries most women in reproductive age are active workers, and most of them safely carry on their working activities during pregnancy until the maternity leave starts. Birth cohorts prospectively collect data and provide a useful resource to investigate the association between maternal work-related risk factors and children's health, including adverse birth outcomes. In addition, these studies can reduce biases inherent in cross-sectional or retrospective studies and have a specific focus on critical periods of exposures, such as pregnancy, that may be overlooked by traditional occupational epidemiologic studies that typically collect occupational and medical histories.

Although some studies based on mother-child cohorts observed no differences in the risk of adverse birth outcomes between employed and women without a paid employment [3,4], a more recent collaborative study involving 13 cohorts in Europe, including 2626 women of the NINFEA cohort, found that working in some occupational sectors may affect the normal reproductive processes of pregnancy and foetal development [5]. Differences in the exposure prevalence of physical and biomechanical factors, biological and chemical agents, psychosocial risks and working hours were indeed observed between different sectors such as manufacturing and construction industries, administrative services, and human health and social work [6]. The magnitude of the effect of working conditions on birth outcomes may vary with the source, level, and timing of exposures in different settings and environments. A systematic review showed that chemical, psychosocial and physical-ergonomic-mechanical factors were the main occupational exposures associated with low birthweight, preterm birth, miscarriage, and other obstetric complications [7]. However, some occupations often do not entail only exposures to a single agent, and the assessment of multiple exposures can be helpful to identify concurrent risks that may impact foetal health, as shown in some population-based studies [8–11].

In this study, we examine a wide range of employment-related exposures in women who participated in the Italian web-based mother-child cohort NINFEA, focusing on employed and nulliparous women. Working conditions and exposures to physical and chemical agents are assessed using both prospectively collected data from questionnaires and Job-Exposure-Matrices (JEMs), focusing on the period during the early phases of pregnancy. We characterise patterns of co-exposures and use birth cohort data to account for several potential confounders assessed at the individual level. Maternal occupational exposures during early pregnancy are then studied in association with child's birthweight, the risk of preterm birth, and caesarean section.

## Methods

### Design and study population

The NINFEA study is a prospective Italian birth cohort that recruited more than 7500 pregnant women (www.progettoninfea.it) [12,13] from 2005 to 2016. Participation in the study was voluntary: all pregnant women who had access to the internet and sufficient knowledge of the Italian language to complete online questionnaires could participate in the study. Enrolled

women completed a baseline online questionnaire during pregnancy that collected information on environmental, lifestyle and health factors. After delivery, mothers were invited to fill online follow-up questionnaires at 6 and 18 months of age of their child, and then when they turned four, seven, ten, thirteen and sixteen years old.

In this study, we have included women who had complete information on employment status and job description (collected through the questionnaire administered during pregnancy) and on birth outcomes (collected from the questionnaire completed at 6 months of the child). All employed mothers were included, whether on permanent or temporary contract, self-employed or freelancer/entrepreneurs. We used information from the first questionnaire administered during pregnancy of the NINFEA database version 2022.02.

## Birth outcomes

We considered three birth outcomes: birthweight (g), preterm birth (<37 gestational weeks), and caesarean delivery. All the outcomes were based on information reported by the mothers either during pregnancy (last menstrual period and gestational age based on ultrasound examination) or in the 6-month questionnaire (birthweight, gestational age at delivery, and mode of delivery). Gestational age was established based on the best judgement between last menstrual period, ultrasound examinations and maternal report, with preterm birth being defined as such for babies born before 37 weeks of pregnancy. Mode of delivery was grouped into any caesarean (emergency, elective, or for unknown reasons) vs. vaginal delivery. Although there are many different determinants of caesarean section, we included it among adverse outcomes as a possible proxy of pregnancy complications that may lead to non-elective caesarean, such as, for instance, foetal distress, placental complications, and pre-eclampsia [14–16].

## Maternal occupation and working conditions

The questionnaire completed during pregnancy included questions about employment status at the beginning of pregnancy, and occupational exposures and working conditions in the first trimester of pregnancy. Considering that the information collected relates to the beginning of pregnancy, and thus also to its early stages when the woman may not yet be aware of the pregnancy, the exposures detected reflect situations in which the necessary precautionary measures required by maternity protection laws may not yet be in place Information on occupation was obtained through three questions: (i) employment status (e.g., permanent contract, temporary contract, freelancer/entrepreneur, self-employed,); (ii) job title (free-text field); and (iii) occupational branch (free-text field). Women without paid employment were not included in this study (unemployed looking for a new occupation, unemployed and looking for a first job, housewife, student). Women's occupations were first classified according to the ISCO-88 (International Standard Classification of Occupations 1988) [17], using mainly the job description, aided by the information on the occupational branch. By applying a crosswalk matrix, ISCO-88 codes were then translated into the CNO-94 codes (Spanish National Classification of Occupations 1994) [18] to enable the subsequent linkage between job titles and selected JEMs.

The CNO-94 classification is based on four levels of job description accuracy, detailed using an increasing number of digits as the accuracy of the description increases, in a manner equivalent to the ISCO classification system.

## JEMs

In this study we used two JEMs. The MatEmEsp [19] is a Spanish JEM covering the period 1996–2005 and including a broad selection of occupational exposures (ergonomic, safety,

hygiene and psychosocial). The second JEM, developed by Van Tongeren and colleagues [20], and previously used in the collaborative study by Birks et al, [21] that included information of 2455 women of the NINFEA cohort, was used to assess exposure to endocrine-disruptor chemicals (EDCs).

The MatEmEsp provides the prevalence of exposure to specific agents for each job. The prevalence of ergonomic risk factors was assigned to jobs defined with CNO-94 three-digit codes whereas physical, biological, and chemical exposures were defined for CNO-94 four-digit codes.

Through linkage between CNO-94 codes and the MatEmEsp, women were classified as exposed if the exposure prevalence derived from the JEM for their job was greater than or equal to 75%. Exposures were included in the study only if at least twenty women were classified as exposed. Heat stress, workload, working in standing position, repetitive movements, sedentary work, video display terminals (VDT), detergents and animal dust were thus included in the analysis.

The JEM developed by Van Tongeren and colleagues was used to assess exposure to at least one endocrine-disruptor chemical among pesticides, polychlorinated organic compounds, phthalates, alkylphenolic compounds, bi-phenolic compounds, heavy metals, and other hormone-disrupting substances (20). EDCs exposure was reported as a binary variable (yes/no) and information was available only for four-digits ISCO-88-coded jobs.

Maternal job codes defined by less than four digits for physical, biological, chemical, and EDCs exposure and by less than three digits for ergonomic risks did not link with the JEMs. Exposures for such job codes were imputed according to the following procedure. Let us consider a job code defined with three digits, for example, CNO-94 code "211" corresponding to "life science professionals", for which we want to define exposure to a specific agent, for example *detergents*, and which is defined in the MatEmEsp only for four-digits codes. We take all job codes in the NINFEA cohort defined by four digits whose first three coincide with those of the one we want to impute, and that link with the JEM. In our example, these are "2111" ("biologists, botanists, zoologists and related professionals"), "2112" ("pharmacologists, pathologists and related professionals"), and "2113" ("agronomists and related professionals"). We then look at the distribution of women in the cohort with jobs classified with any of the above four-digit codes: if at least 75% of them are classified as exposed to the agent of interest, then women with a job classified with only the first same three initial digits are classified as exposed. In our example, women with the "211" job would be classified as exposed if at least 75% of women of our cohort with jobs "2111", "2112" or "2113" are exposed to *detergents*.

## Questionnaire

In the questionnaire, women answered questions about working load perception (including demanding work, repetitive work, energy expenditure, and stress), reduction of working hours, night shifts and noise. A checklist was used to obtain information on work-related chemical exposures, including diesel engine exhaust (DEE), dust, inks, oils, solvents, paints, chemotherapeutics, formalin, and anaesthetic gases. Work-related passive smoking was included in the group of chemicals assessed.

When information on exposure to DEE, solvents, noise, and night shifts were available in both the MatEmEsp and the questionnaire, we decided to prioritise the information on exposures from the questionnaire. This decision was made because JEMs are not tailored for women of childbearing age and assume average levels of exposure for each job, unlike the questionnaires, which provide exposure information at the individual level and are not prone to Berkson type error.

All exposure variables were reported as binary variables (yes/no). Questions collected from the questionnaire, all available answers and transformation of variables from categorical to binary are described S1 Table in S1 File. Analogously to the exposures obtained using the JEMs, also exposures derived from questionnaire were included in the analysis only if at least twenty women were exposed.

## Confounders

Information on maternal age at child's birth, smoking during pregnancy (yes or no), country of birth (Italy or abroad), pre-pregnancy weight and marital status, and on the child's region of birth (Piedmont, Tuscany, other northern Italian regions, other Italian regions) was collected from the questionnaire and included in the analysis as covariates potentially influencing participation in the NINFEA study or potential confounders that were selected because common causes of both the exposure and the outcome, guided by a priori knowledge.

As previous pregnancies and their outcomes may have strong and insidious confounding effects in studies of occupational exposures in pregnancy, especially when focusing on birth outcomes, we restricted the analyses to nulliparous women.

## Statistical analysis

We used multivariable linear regression models for continuous outcome variables (birthweight) and multivariable logistic regression models for dichotomous outcome variables (preterm birth and caesarean delivery). Each exposure-outcome association was estimated separately. In addition, we carried out an Exploratory Factor Analysis (EFA) [22] to characterise patterns of co-exposures and to summarise this information into a small set of latent constructs (factors). The optimal number of factors in the EFA analysis was found to be six according to the Very Simple Structure criterion [23,24]. The strength of the relationship between factors and each exposure is described via factor loadings that can be interpreted similarly to correlation coefficients, with values ranging from -1 to 1. Individual scores, representing an individual's placement on the factor(s), were computed by means of the Thurstone method [25]. The outcomes of interest were then regressed on the individuals' factor scores using linear and logistic regression models, as described above.

A sensitivity analysis was carried out considering only women with jobs defined with complete information at four digits to evaluate the impact of imputation. A further sensitivity analysis was performed to evaluate differences in exposure assignment using imputation from the JEMs at a 50% threshold for the prevalence of exposure rather than the used 75% threshold.

## Ethics

Ethical approval was obtained from the Azienda Sanitaria Ospedaliera "San Giovanni Battista" of Turin Ethics Committee. Mothers gave written informed consent at enrolment.

## Results

### Description of the study population

Table 1 shows the characteristics of the study population of pregnant women and the distribution of birth outcomes. The number of mothers included in this study was 3938 after the exclusion of women without paid employment (n = 888) and multiparous women (n = 2006), women with missing information on their job status (n = 1049) and those who registered into the study after the 37th week of gestation (n = 228). Their mean age at conception was 32.8 years, ranging from 19 to 50 years. Most women (97%) were born in Italy. The average

**Table 1. Characteristics of pregnant women (n = 3938) of the NINFEA birth cohort included in the study.**

| Variables | n | % |
|---|---|---|
| *Individual characteristics* | | |
| Age at enrolment in study | | |
| < 25 years | 64 | 2 |
| 25–29 years | 476 | 12 |
| 30–35 | 2106 | 54 |
| > 35 years | 1292 | 32 |
| *Country of origin* | | |
| Italy | 3816 | 97 |
| Abroad | 122 | 3 |
| *Cohabitation with partner during pregnancy* | | |
| Yes | 3281 | 93 |
| No | 269 | 7 |
| *Smoking during pregnancy* | | |
| Yes | 324 | 8 |
| No | 3938 | 92 |
| *Employment position* | | |
| Business and public administration | 188 | 5 |
| Scientific and technicians, and professionals | 1322 | 33 |
| Support technicians and professionals | 492 | 12 |
| Administrative-type employees | 1528 | 39 |
| Service workers and shop and market sales workers | 309 | 8 |
| Skilled agricultural and fishery workers | 6 | < 1 |
| Craft and related trades workers | 73 | 2 |
| Plant and machine operators and assemblers | 8 | < 1 |
| Elementary occupations | 10 | < 1 |
| *Birth outcomes* | | |
| Preterm birth (n = 3937) | | |
| Yes | 266 | 7 |
| No | 3671 | 93 |
| Caesarean delivery (n = 3752) | | |
| Yes | 1053 | 28 |
| No | 2699 | 72 |
| Birthweight, grams (n = 3833); (mean = 3203, SD = 505) | | |

birthweight of their children was 3203 g (SD 505 g) and 27% of deliveries were caesarean sections. The prevalence of low birthweight (< 2500 g) was 7%, and approximately 7% of children were born preterm.

With regards to their occupation, NINFEA mothers were administrative-type employees (39%), scientific technicians and professionals, and intellectuals (33%), support technicians and professionals (12%), service workers and shop and market sales workers (8%), business and public administration workers (5%), craftspeople and workers qualified for the manufacturing, construction, and mining industries (2%), operators and fitters of fixed machinery, and drivers and operators of mobile machinery (0.2%), workers in elementary occupations (0.6%) and skilled agriculture and fishing workers (0.2%) (S2 Table in S1 File). Tables 2 and 3 show the number of working mothers by exposure group and agents based on information from the questionnaire and the JEMs, respectively.

**Table 2. Occupational exposure groups, agents, and number of exposed NINFEA women derived from the questionnaire.**

| Exposure group | Agents | N of exposed (%) |
|---|---|---|
| Chemicals agents | Dust | 19 (< 1) |
| | Chemotherapeutics | 19 (< 1) |
| | Formalin | 27 (< 1) |
| | DEE | 29 (< 1) |
| | Gas Anaesthetics | 39 (1) |
| | Inks | 49 (1) |
| | Oils | 17 (<1) |
| | Solvents | 77 (2) |
| | Paints | 27 (< 1) |
| | Passive smoking | 164 (4) |
| Physical agents | Noise | 909 (23) |
| Working load perception | Energy expenditure | 673 (17) |
| | Repetitive work | 974 (25) |
| | Demanding work | 2016 (52) |
| | Stress | 491 (12) |
| Shift | Night shift | 187 (5) |
| | Reduction working hours | 1556 (39) |

Abbreviations: DEE = Diesel Engine Exhaust.

## Occupational exposure from questionnaire and birth outcomes

Table 4 shows the associations of working exposures with birthweight, preterm birth, and caesarean delivery.

Mothers who reported to have reduced their working hours during the first trimester of pregnancy gave birth to children with lower birthweight—43.0 g (95% CI (Confidence Interval)– 76.4 to -9.7) and had increased odds ratio (OR) of preterm birth (OR 1.52 95% CI 1.15 to 2.00), and caesarean delivery (OR 1.27 95% CI 1.09 to 1.49) compared to women who stated that their working hours remained unchanged. Demanding work was associated with a 40.4 g (95% CI 7.5 to 73.4) increase in children birthweight and decreased odds of preterm birth (OR 0.72 95% CI 0.54 to 0.95).

**Table 3. Exposure groups, agents, and number of exposed NINFEA women derived from the MatEmEsp and the "Endocrine Disruptor Chemicals" JEM.**

| Exposure group | Agents | N of exposed (%) |
|---|---|---|
| Chemicals agents | Detergents | 265 (7) |
| | Endocrine disruptors chemicals | 151 (4) |
| Biological agents | Animal dust | 34 (<1) |
| | Heat | 56 (1) |
| Physical agents | VDT | 1749 (44) |
| Working load | Workload | 47 (1) |
| | Standing work | 43 (1) |
| | Sedentary work | 1696 (43) |
| | Repetitive movements | 76 (2) |

Abbreviations: VDT = Video display terminal.

**Table 4. Multivariable linear and logistic regressions of the associations of occupational exposures of women (n = 3938) during the first trimester of pregnancy with birthweight, preterm birth (< 37 weeks), and caesarean delivery.**

| Occupational exposure | Birthweight | Pre-term birth | Caesarean section |
|---|---|---|---|
| | ß (95% CI) | OR (95% CI) [n. cases exposed] | OR (95% CI) [n. cases exposed] |
| Questionnaire/self-reported exposure | | | |
| Dust | 61.4 (-186.9 to 309.7) | 2.13 (0.34 to 8.10) [2] | 6.99(2.36 to 25.47) [12] |
| Chemotherapeutics | 78.4 (-148.1 to 304.9) | / | 0.60(0.13 to 1.88) [4] |
| Formalin | 37.7 (-154.8 to 230.3) | 1.24 (0.20 to 4.28) [2] | 1.58 (0.66 to 3.62) [10] |
| DEE | -68.5 (-250.6 to 113.4) | 1.15 (0.18 to 3.91) [3] | 1.72 (0.75 to 3.74) [11] |
| Gas Anaesthetics | -37.0 (-195.7 to 121.7) | 1.17 (0.28 to 3.34) [3] | 1.46 (0.70 to 2.94) [14] |
| Inks | 17.4 (-139.2 to 174.0) | 1.36 (0.32 to 3.87) [5] | 1.63 (0.80 to 3.19) [11] |
| Oils | 160.5 (-72.7 to 393.8) | / | 1.41 (0.44 to 3.97) [5] |
| Solvents | 55.41 (-64.7 to 175.5) | 0.23 (0.01 to 1.05) [3] | 0.67 (0.34 to 1.24) [14] |
| Paints | 11.3 (-203.8 to 175.5) | 0.75 (0.04 to 3.68) [1] | 0.49 (0.11 to 1.49) [5] |
| Passive smoking | -72.4 (-158.8 to 14.0) | 1.94 (1.04 to 3.35) [19] | 1.59 (1.08 to 2.32) [64] |
| Noise | -15.7 (-57.4 to 26.0) | 0.93 (0.65 to 1.33) [54] | 1.03 (0.85 to 1.26) [247] |
| Reduction working hours | -43.1 (-76.4 to -9.7) | 1.52 (1.15 to 2.00) [130] | 1.27 (1.09 to 1.49) [470] |
| Stress | 6.2 (-45.3 to 57.7) | 1.04 (0.66 to 1.60) [32] | 1.08 (0.85 to 1.37) [138] |
| Energy expenditure | 36.1 (-6.7 to 78.9) | 0.76 (0.51 to 1.10) [40] | 1.08 (0.88 to 1.31) [321] |
| Repetitive work | -47.5 (-86.2 to -8.9) | 0.91 (0.65 to 1.26) [54] | 0.91 (0.75 to 1.09) [229] |
| Demanding work | 40.5 (7.5 to 73.4) | 0.72 (0.54 to 0.94) [120] | 0.89 (0.76 to 1.03) [512] |

(*Continued*)

**Table 4.** (Continued)

| Occupational exposure | Birthweight | Pre-term birth | Caesarean section |
|---|---|---|---|
| Night shift | 14.4<br>(-60.7 to 89.6) | 0.62<br>(0.26 to 1.25)<br>[10] | 1.07<br>(0.75 to 1.51)<br>[51] |
| Job-Exposure Matrix | | | |
| Heat | -160.2<br>(-299.6 to -20.7) | 1.47<br>(0.44 to 3.72)<br>[4] | 1.19<br>(0.60 to 2.25)<br>[15] |
| Workload | 65.2<br>(-90.1 to 220.5) | /<br>[3] | 1.73<br>(0.86 to 3.35)<br>[15] |
| Standing work | -72.8<br>(-225.8 to 80.2) | 2.15<br>(0.63 to 5.54)<br>[4] | 0.53<br>(0.18 to 1.27)<br>[5] |
| VDT | -18.5<br>(-51.3 to 14.3) | 1.12<br>(0.85 to 1.47)<br>[125] | 1.11<br>(0.94 to 1.29)<br>[479] |
| Sedentary | -22.3<br>(-55.2 to 10.6) | 1.14<br>(0.87 to 1.50)<br>[122] | 1.11<br>(0.95 to 1.29)<br>[466] |
| Repetitive movements | -55.7<br>(-174.5 to 62.9) | 1.75<br>(0.67 to 3.84)<br>[6] | 0.58<br>(0.27 to 1.11)<br>[12] |
| Detergents | -13.7<br>(-78.6 to 51.2) | 0.63<br>(0.30 to 1.19)<br>[14] | 1.25<br>(0.92 to 1.69)<br>[74] |
| Animal dust | 62.3<br>(-108.2 to 232.7) | / | 1.03<br>(0.44 to 2.19)<br>[10] |
| Endocrine Disruptors | -78.8<br>(-164.2 to 6.6) | 1.45<br>(0.70 to 2.70)<br>[12] | 1.32<br>(0.88 to 1.96)<br>[42] |
| Factor analysis | | | |
| Healthcare chemicals and active work | -14.9<br>(-4.4 to 34.2) | 0.91<br>(0.70 to 1.10) | 0.90<br>(0.80 to 0.99) |
| Industrial chemicals | 3.2<br>(-15.3 to 21.7) | 1.06<br>(0.89 to 1.20) | 1.03<br>(0.94 to 1.12) |
| Physical-work and EDCs | 14.2<br>(-5.9 to 34.2) | 0.78<br>(0.56 to 1.02) | 0.94<br>(0.84 to 1.04) |
| Outdoor work | 4.4<br>(-21.0 to 29.8) | 1.12<br>(0.90 to 1.34) | 0.97<br>(0.86 to 1.10) |
| Workload and detergents | 1.4<br>(-16.2 to 18.9) | 0.90<br>(0.72 to 1.07) | 1.02<br>(0.94 to 1.10) |
| Stressful work | 18.6<br>(-0.1 to 37.3) | 0.89<br>(0.74 to 1.08) | 1.01<br>(0.93 to 1.11) |

**Abbreviations:** N = number of subjects; DEE = Diesel Engine Exhaust; VDT = Video Display Terminal; n. cases exposed: number of exposed with the outcome.

Work-related passive smoking exposure was associated with a lower birthweight (β = -72.3 95% CI -158.8 to 14.0) and with an increased OR of preterm birth (OR 1.94 95% CI 1.04 to 3.35), and caesarean section (1.59 OR 95% CI 1.08 to 2.32). Repetitive work was associated with a lower birthweight (β = -43.0 95% CI -76.4 to -9.7).

A high OR of caesarean section (OR 6.99 95% CI 2.36 to 25.47) was observed for women exposed to dust.

There was a suggestive association of decreased birthweight (β = -68.5 95% CI -250.6 to 113.4) with maternal exposure to DEE. Decreased odds of preterm birth were apparent for children born to mothers exposed to solvents (OR 0.23 95% CI 0.01 to 1.05) and chemotherapeutics (OR 0.47 95% CI 0.10 to 1.46).

## JEMs and birth outcomes

Exposure to heat was associated with a lower birthweight (β = -160.1 95% CI -299.6 to -20.7) and an increased OR of preterm birth (OR 1.47 95% CI 0.44 to 3.72). Working in a standing position was associated with an elevated OR of preterm birth (OR 2.15 95% CI 0.63 to 5.54) but with a decreased OR of caesarean delivery (OR 0.53 95% CI 0.18 to 1.27). Similarly, the OR of preterm birth (OR 1.75 95% CI 0.67 to 3.84) was higher and that of caesarean delivery was lower (OR 0.58 95% CI 0.27 to 1.11) for exposure to repetitive movements.

The sensitivity analysis (S3 Table in S1 File) that included only mothers with jobs specified with 4 digits (n = 1986) in the CNO-94 classification system did not considerably change the estimated associations except for the exposure to heat. The sensitivity analysis carried out to test a different threshold (i.e., 50%) of exposure imputation had a stronger impact and sometimes yielded exposure-outcome associations in the opposite direction (S4 Table in S1 File).

## Factor analysis

The proportion of variance explained by the six selected factors in the EFA accounted for 60% of the total variance. Fig 1 shows the factors, and the standardised loadings based on the correlation matrix between factors and exposures. "Healthcare chemicals and active-work" factor was loaded by formalin (r = 0.79), anaesthetics gases (r = 0.76) and chemotherapeutics (r = 0.64) whereas an inverse correlation was observed for VDT (r = -0.50) and working in a sedentary position (r = -0.49). Oils (r = 0.85), paints (r = 0.77), solvents (r = 0.75) and DEE (r = 0.72) determined the loading of factor "Industrial chemicals". Working in a standing position (r = 0.96), repetitive movements (r = 0.80) and EDCs (r = 0.65) loaded factor "Physical-work and EDCs", while heat (r = 0.83) and animal dust (r = 0.71) loaded factor "Outdoor work". Factor "Workload and detergents" was loaded by workload (r = 0.76) and detergents (r = 0.75), and inversely correlated to VDT (r = -0.53) and sedentary work (r = -0.52). The factor "Stressful work" was loaded by energy expenditure (r = 0.84) and stress (r = 0.70).

Overall, no associations were found between the factors and pregnancy outcomes except for the factor "Healthcare chemicals and active-work", which indicated a decrease in the odds of caesarean section (OR = 0.90, 95% CI 0.80 to 0.99) and the factor "Stressful work", which showed an increase in birthweight (ß = 18.6, 95% CI -0.4 to 37.3).

## Discussion

In the Italian NINFEA population-based birth cohort study, exposure to demanding work was associated with increased birthweight and decreased odds ratio of preterm delivery. Studies have shown that women who are better able to perform physically demanding tasks need good physical health [26]. This may partly explain our observed associations, since good health, being a determinant of both pregnancy outcomes and being at work, could give rise to a healthy worker effect bias [27,28].

The association between self-reported reduced working hours during early pregnancy and adverse birth outcomes is likely explained by reverse causality. Women that had a high-risk

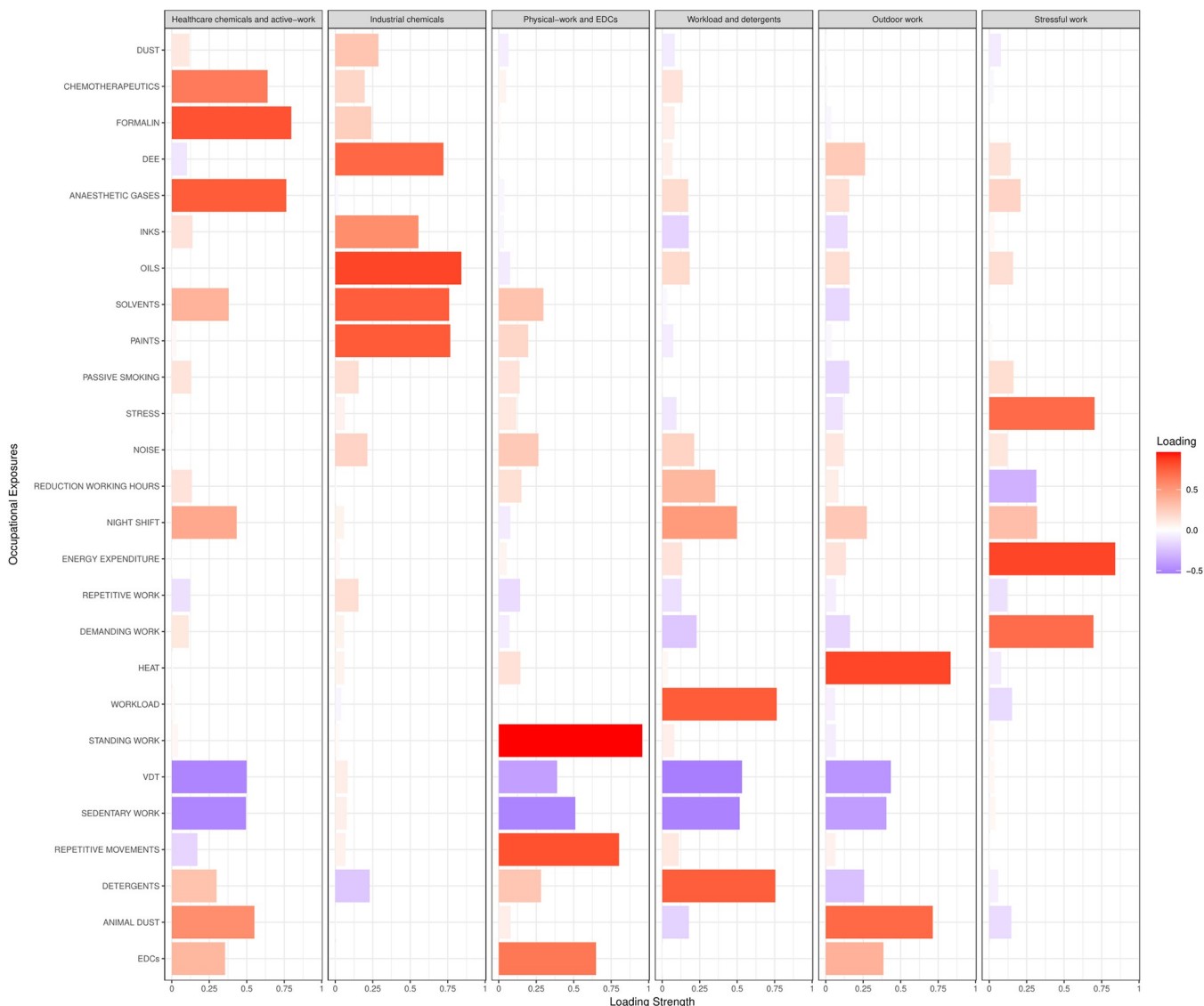

**Fig 1. Exploratory factor analysis (EFA) (17) to characterise patterns of occupational co-exposures of NINFEA women.** The six identified factors are shown with their loading in terms of maternal occupational exposures: Positive (red) and negative (blue) correlations between factor and exposure. *Abbreviations:* *DEE = Diesel Engine Exhaust; VDT = Video Display Terminal; EDCs = Endocrine Disruptor Chemicals.*

pregnancy, potentially associated with adverse birth outcomes, may have had to reduce their workload when the high-risk pregnancy condition became apparent.

For some occupational exposures, we found associations with birth outcomes in line with the current literature. In two large studies, Norlen and colleagues found associations between exposure to organic and inorganic dust among full-time working mothers with low absence from work and preterm birth, the birth of children small for gestational age, and low birth-weight [29,30]. In our study, the prevalence of exposure to dust was low, resulting in a small number of exposed women and wide confidence intervals. Furthermore, the composition of dust was unknown.

We observed an increased risk of all adverse birth outcomes in association with work-related passive smoking in line with the main findings of a review estimating that women

exposed to environmental passive smoking were 1.5–4 times more likely to deliver a child with low birthweight [31], even if our study focused only on occupational exposure. Although the law in Italy bans smoking indoors, 4% of women reported to be exposed to passive smoking during their working shifts.

More than 5% of women between 15–49 years old work at night in Italy [32]. A large review summarising 38 meta-analyses and 24 systematic reviews showed a link between night shift work and insufficient sleep, accidents, weight gain and coronary and cardiometabolic diseases [33]. However, our results showed that women working night shifts experienced a decreased risk of all adverse birth outcomes. Comparable results were found in a study showing that the relative risks of overweight or obesity were modestly lower in children with mothers working night shifts before pregnancy compared to children with mothers that stopped working night shifts during pregnancy [34]. As reported above with regard to women that reduced their working hours during the first trimester of pregnancy, it may be plausible that women with better physical conditions and without fertility problems are more likely to continue night shifts during the early phases of pregnancy.

Rising mean temperatures and increasing frequency and intensity of heatwaves due to climate change pose particular attention on susceptibility and vulnerability of pregnant women, especially in workplaces where physical activity can be more intense. In our study, maternal exposure to heat, which was defined through the MatEmEsp with the ISO standard 7243, was associated with adverse birth outcomes. To our knowledge, no studies have been conducted in Europe exploring the impact of heat stress in the workplace on birth outcomes. However, a systematic review and a meta-analysis, showed that environmental exposure to 1°C increase in temperature was associated with 5% increased risk of preterm birth and stillbirths [35].

"One exposure-one disease" is a common approach also used in settings of occupational studies to analyse a single level of exposure in association with a single health endpoint. In this study, we used an exploratory factor analysis to characterise maternal occupational exposure profiles during pregnancy and investigate the relationships between patterns of co-exposures and birth outcomes. The first identified factor seems to capture several co-exposures frequent among healthcare workers, who are numerous in the cohort. The factor "Industrial chemicals" profiled co-exposures of women to different macro groups of chemicals, suggesting the presence of jobs often involving simultaneous exposure to multiple chemicals rather than to individual ones. Standing work, repetitive movements, and EDCs were highly correlated within the third factor that, for this reason, was named "Physical-work and EDCs". The factor "Outdoor work" showed co-exposure to heat and animal dust plausibly related to agricultural workers, and the factor "Workload and detergents" showed a co-exposure between workload and detergents. Interlinked and complementary elements of exposure as stress, energy expenditure and demanding work were present in the factor "Stressful work". Overall, no associations were observed in models where birth outcomes were regressed on individual scores obtained from the exploratory factor analysis. The weak estimated associations could be caused by a stronger effect of single exposures than that of co-exposures considered together, although also the associations estimated analysing single agents were somehow weak. Furthermore, women exposure to several agents and working conditions might have acted in opposite directions.

The large sample size of nulliparous women, the prospective design and collection of information about confounders and birth outcomes, and the use of two distinct exposure assessment methods (questionnaires and JEMs) are the main strengths of this study.

In addition to the potential to reduce biases, the importance of prospective birth cohort studies in occupational epidemiology lies in the application of a life-course conceptual model to a broader context of occupational health, aiming at investigating how mothers' working life may influence their current and future health status, and how the interaction between their

working life and health may impact their children's health over time [36]. Early life multiple exposures have been shown to be crucial for children's health [37] but little is known about the interplay between work, parents' and children's health across different labour markets and social contexts. This population-based study provided an ideal framework to identify multiple risks for working women during a highly susceptible period of life, with critical implications for children's health.

With regards to the limitations of this study, the heterogeneity of occupations represented in the NINFEA cohort was small, with a large number of clerks and healthcare workers, a high prevalence of mothers with medium-high educational level [38], and low enrolment below age 25 and minority groups. Low prevalence of exposures to chemical and physical agents may be explained by the low number of women engaged in activities requiring lower and intermediate skills. Consequently, the occupations of NINFEA's mothers implied a high prevalence of sedentary work and VDT exposure that were not found to be associated with birth outcomes, as already observed in previous studies [39,40]. Because of the low prevalence of exposure among NINFEA mothers, many associations were estimated with low precision.

To minimise excessive reliance on p-values, we chose not to adjust for multiple comparisons and to focus, instead, on analysing result patterns [41].

In summary, exposure to demanding work was associated with a reduced risk of adverse birth outcomes perhaps partly explained by the healthy worker effect. Maternal exposure to passive smoking, dust and heat were associated with an increased risk of adverse outcomes. A reduction of working hours and night shifts were associated with an increased and decreased risk of adverse outcomes, respectively, possibly explained by reverse causality and healthy worker effect. Multiple and co-exposures were assessed by exploratory factor analysis to document concomitant occupational risks among women without showing strong associations with birth outcomes.

## Supporting information

**S1 File.**
(DOCX)

## Acknowledgments

The authors are grateful to all the participants of the NINFEA cohort. We thank Dr. Ana Maria Garcia to provide us the job-exposure matrix "MatEmESp" for this study. This work was partly supported by the Compagnia di San Paolo, 'Bando per il finanziamento Ex-Post di Progetti di Ricerca di Ateneo–Anno 2020.

## Author Contributions

**Conceptualization:** Antonio d'Errico, Maja Popovic, Lorenzo Richiardi, Milena Maule.

**Data curation:** Antonio d'Errico, Maja Popovic, Milena Maule.

**Formal analysis:** Antonio d'Errico, Lorenzo Richiardi, Milena Maule.

**Funding acquisition:** Lorenzo Richiardi.

**Investigation:** Antonio d'Errico, Maja Popovic, Giovenale Moirano, Lorenzo Richiardi, Milena Maule.

**Methodology:** Lorenzo Richiardi, Milena Maule.

**Project administration:** Antonio d'Errico, Lorenzo Richiardi, Milena Maule.

**Resources:** Antonio d'Errico, Maja Popovic.

**Software:** Antonio d'Errico.

**Supervision:** Maja Popovic, Costanza Pizzi, Giovenale Moirano, Lorenzo Richiardi, Milena Maule.

**Validation:** Antonio d'Errico, Maja Popovic.

**Visualization:** Antonio d'Errico, Giovenale Moirano.

**Writing – original draft:** Antonio d'Errico, Milena Maule.

**Writing – review & editing:** Antonio d'Errico, Maja Popovic, Costanza Pizzi, Giovenale Moirano, Chiara Moccia, Lorenzo Richiardi, Milena Maule.

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
