## [Decision Letter · Decision Letter 0]

23 Jan 2024

PONE-D-23-32546Maternal occupational exposures in the NINFEA birth-cohort during the pre-conception period and early stages of pregnancyPLOS ONE

Dear Dr. d'Errico,

Thank you for submitting your manuscript to PLOS ONE. After careful consideration, we feel that it has merit but does not fully meet PLOS ONE’s publication criteria as it currently stands. Therefore, we invite you to submit a revised version of the manuscript that addresses the points raised during the review process.

In particular, please address reviewer 2's comments about distinguishing the prenatal and preconception periods, both in the methods and results of your study, and in the broader context. Our apologies for the slowness of the reviews.==============================

We look forward to receiving your revised manuscript.

Kind regards,

Emily W. Harville

Academic Editor

PLOS ONE

2. In the online submission form, you indicated that Data are not publicly available. Data from this study are available upon reasonable request from Lorenzo Richiardi and are subject to local, national, and European rules and regulations.

Reviewers' comments:

Reviewer's Responses to Questions

**Comments to the Author**

1. Is the manuscript technically sound, and do the data support the conclusions?

Reviewer #1: Yes

Reviewer #2: Partly

2. Has the statistical analysis been performed appropriately and rigorously? 

Reviewer #1: Yes

Reviewer #2: Yes

3. Have the authors made all data underlying the findings in their manuscript fully available?

Reviewer #1: Yes

Reviewer #2: No

4. Is the manuscript presented in an intelligible fashion and written in standard English?

Reviewer #1: Yes

Reviewer #2: Yes

5. Review Comments to the Author

Reviewer #1: The manuscript is a valuable addition to the existing literature investigating the association of maternal environmental exposures before and in the early stages of pregnancy with adverse birth outcomes. The authors have also considered a wide range of maternal occupational exposures. However, in the manuscript, it is unclear why maternal education was not considered a potential confounding factor since maternal education could be associated with adverse birth outcomes, especially with delivery by cesarean section. The association between reduced working hours during early pregnancy and adverse birth outcomes is interesting, and the authors explain it by reverse causality in the discussion section. It is mentioned that high-risk pregnancy could be potentially associated with adverse reproductive outcomes. If data on adverse birth outcomes (e.g., history of abortion) is available, accounting for this factor could better explain the findings.

Reviewer #2: General and major comments

The topic of the manuscript is relevant, there is not a tremendous amount of studies out there on occupational exposure pre-conception and during pregnancy and negative birth outcomes, and more studies, especially on multiple exposures are of relevance. The paper is descriptive in nature and present a lot of analysis, but is still informative. There is, however, some basic issues with the current version of the manuscript. Most importantly, exposure both preconception and during early pregnancy is mentioned in the title (surprisingly the outcomes are not) but the fundamental differences related to these two exposure windows are not mentioned at all in the manuscript, and it is unclear why a few pre-conception exposures are included. The is no distinction between pre-conception and during pregnancy in the results presented in the abstract. If the pre-conception approach should be kept in the manuscript I propose to provide much more information about this in the introduction, the interpretation of the results and the discussion.

With regards to the outcomes, I am not sure the composite variable adds a lot. i.e. it merges very different outcomes, especially caesarean section vs. the two other outcomes. I suggest the authors to stick to the 3 original outcomes (birthweight, preterm birth, caesarean section). Did the authors also considered to include small for gestational age?

Page 6, line 151-154. It seems like the questionnaire information was preferred at the expense of JEM information, please make this more explicit in the text. Did you consider to analyse similar exposures from both exposure sources?

Page 6, line 160 – 166. Please state the rationale for the chosen confounder to be included.

Page 12, line 332.333. The authors argue against adjusting for multiple comparisons, and state they focus on patterns, but this is not very clear to me, and the pattern approach should be more explicitly described and discussed. Abd still, they perform 148 analysis, and it is highly plausible some of the significant findings are random findings.

Minor comments

Page 5, line 130 – 141: The description of the imputation can be substantially condensed.

6. PLOS authors have the option to publish the peer review history of their article (what does this mean?). If published, this will include your full peer review and any attached files.

Reviewer #1: No

Reviewer #2: **Yes: **Vivi Schlünssen

---

## [Author Response · Author response to Decision Letter 0]

12 Apr 2024

Reviewer #1: The manuscript is a valuable addition to the existing literature investigating the association of maternal environmental exposures before and in the early stages of pregnancy with adverse birth outcomes. The authors have also considered a wide range of maternal occupational exposures. However, in the manuscript, it is unclear why maternal education was not considered a potential confounding factor since maternal education could be associated with adverse birth outcomes, especially with delivery by cesarean section. The association between reduced working hours during early pregnancy and adverse birth outcomes is interesting, and the authors explain it by reverse causality in the discussion section. It is mentioned that high-risk pregnancy could be potentially associated with adverse reproductive outcomes. If data on adverse birth outcomes (e.g., history of abortion) is available, accounting for this factor could better explain the findings.

Answer #1: 

Thank you for revising our work and providing your insightful comments. 

1. Regarding your understandable concern about maternal education not being considered as a confounder in our study, it is important to note that this exclusion stems from an ongoing debate within the field of occupational observational studies. Occupational status is frequently used as core component of the socioeconomic status (SES) construct that comprises a complex bundle of correlated factors/mediators such as occupation, material conditions (access to goods and services), psychosocial dynamics and behavioral factors whose impact on health. For instance, in a study examining social inequalities and self-reported health, it was shown that, collectively, these mediators accounted for 78-100% of social differences in health outcomes for both men and women, with occupational factors playing a substantial role (such as rewards, hazardous working conditions, economic activity, decision latitude, and job insecurity) (Aldabe et al., 2012). 

Interestingly, in industry-based studies, the SES comparison becomes more straightforward when analysing jobs where the correlation between job grade and income is weak. This scenario can occur when total income is predominantly influenced by employees' working overtime preferences. For example, in a study involving 15,000 employees across different states in the US, differences in hypertension risks were observed between full-time production workers (blue-collar workers) and production supervisors/administrative staff (white-collar workers). Notably, the hypertension risks were higher among full-time production workers, underscoring the intricate relationship between occupational factors and health outcomes (Hoven et al., 2012).

2. Regarding your question about considering past adverse birth outcomes, such as history of abortion, the analysis were restricted to primiparous women (pregnant for the first time or has been pregnant once). This was reported in the manuscript. 

Reviewer #2: General and major comments

The topic of the manuscript is relevant, there is not a tremendous amount of studies out there on occupational exposure pre-conception and during pregnancy and negative birth outcomes, and more studies, especially on multiple exposures are of relevance. The paper is descriptive in nature and present a lot of analysis, but is still informative. There is, however, some basic issues with the current version of the manuscript. 

Most importantly, exposure both preconception and during early pregnancy is mentioned in the title (surprisingly the outcomes are not) but the fundamental differences related to these two exposure windows are not mentioned at all in the manuscript, and it is unclear why a few pre-conception exposures are included. The is no distinction between pre-conception and during pregnancy in the results presented in the abstract. If the pre-conception approach should be kept in the manuscript I propose to provide much more information about this in the introduction, the interpretation of the results and the discussion.

Thanks for this consideration and to have stressed this limit. We completely agree with you, there was not a solid reason to keep the pre-conception period in the manuscript. Under your suggestion, we considered to remove it.

With regards to the outcomes, I am not sure the composite variable adds a lot. i.e. it merges very different outcomes, especially caesarean section vs. the two other outcomes. I suggest the authors to stick to the 3 original outcomes (birthweight, preterm birth, caesarean section). Did the authors also considered to include small for gestational age?

We opted to retain the three individual outcomes like you were suggesting eliminating the composite one from the manuscript. Regarding "small for gestational age" (SGA), although it was included as an outcome in the initial analysis, its widespread usage lacks a consistent definition, leading to uncertainty and limited utility as a category for identifying at-risk infants, convinced us to not include it as a outcome in our analysis. However, when used alongside preterm birth, SGA can serve as a valuable discriminator, or when gestational age is considered a confounder in predicting mortality based on birthweight (Wilcox et al., 2021).

Page 6, line 151-154. It seems like the questionnaire information was preferred at the expense of JEM information, please make this more explicit in the text. Did you consider to analyse similar exposures from both exposure sources?

1. JEMs are frequently used in situations where assessing occupational exposures proves challenging or when there is limited information available, particularly within high-risk populations (Descatha et al., 2022). While one might assume this applies to our scenario, the NINFEA questionnaire detects a broad range of occupational exposures, albeit with some limitations, which were complemented by the JEMs when appropriate. Our study showed a high prevalence of healthcare workers and clerks, wherein JEMs demonstrated robust performance, particularly in evaluating exposures (often chemicals) within retrospective studies targeting chronic conditions. Moreover, certain occupational risks, such as night-shift work and stress, may be less effectively captured, necessitating a careful evaluation of JEMs performance. Despite the potential for inaccuracies in self-reported exposures, our study minimizes the risk of bias stemming from current or past health conditions, as questionnaires were administered before delivery. Also, exposure variability within jobs (i.e., between workers) can be accounted for when exposure information is collected from the questionnaire, whereas JEMs are prone to the Berkson Error. Taking into account the reasons outlined in our manuscript, we believe that information from the NINFEA-birth cohort questionnaire could be considered more accurate than JEMs in this study context.

2. We conducted Cohen's kappa analysis to evaluate the agreement in exposure assessment between JEMs and the questionnaire for DEE, solvents, noise, and night-shifts. The results indicated very low agreement. While this aspect is not the primary focus of our study, it certainly warrants consideration for future research. Performing a comparison of exposure assessment methods, similar to the approach taken by Ngabirano et al., 2020, specifically tailored to birth cohort data, could offer valuable insights and it can be experimented in an another study. It is a very good idea.

Page 6, line 160 – 166. Please state the rationale for the chosen confounder to be included.

Yes, the sentence included in the manuscript is: potential confounders that were selected because common causes of both the exposure and the outcome, guided by a priori knowledge of their nature. (VanderWeele et al., 2019)

Page 12, line 332.333. The authors argue against adjusting for multiple comparisons, and state they focus on patterns, but this is not very clear to me, and the pattern approach should be more explicitly described and discussed. Abd still, they perform 148 analysis, and it is highly plausible some of the significant findings are random findings.

Your comment underscores the importance of the ongoing discourse in the epidemiological-and other fields, prompting a deeper examination of a critical issue. Approximately 100 or more, mirroring our own situation (148 tests)—the likelihood of at least one incorrect rejection of the null hypothesis (Ho) becomes nearly unavoidable. This finding, as depicted in the figure below, illuminates the intricacies particularly when operating at a significance level context. Moreover, this complexity is compounded when considering correlations between hypotheses and the possibility that not all hypotheses are true. In such scenarios, evaluating error rates becomes a more nuanced task. Also, study design, selected confounders, uncontrolled and unnoticed biases, and research hypothesis, could have affected our (and not only) results.

Figure from Chen et al., 2017 

However, there is not a strict role of choosing to adjust for multiple comparison in relation with our number of tests. My attempt to deepening into this explorative discussion between scientists (epidemiologists and not) can be, of course, limited. Reading some influential works of Greenland and colleagues (Gelman et al., 2012; Greenland et al., 2016; Greenland et al., 2019), can help to light the decision making process. Indeed, the choice to adjust for multiple comparisons often hinges on the specific context and objectives of the inquiry. One broad multiple comparison (MC) context is an exploratory screening (“fishing expedition”), which targets decisions about which associations to study further, reckoning with costs for false leads and missed opportunities. A different broad MC context is simultaneous estimation, which targets accurate summarization of the total information about an entire ensemble of associations, reckoning with trade-offs of bias and random error. Consider, for instance, the intricacies surrounding P-values. P value may be very small because the targeted hypothesis is false; but it may instead (or in addition) be very small because the study protocols were violated, or because it was selected for presentation based on its small size. The general definition of a P value/confidence intervals may help to understand why statistical tests tell us much less than what many think they do. Not only does a P value not tell us whether the hypothesis targeted for testing is true or not; it says nothing specifically related to that hypothesis unless we can be completely assured that every other assumption used for its computation is correct—an assurance that is lacking in far too many studies. 

Our intention was different (even all possible limits of the study) and to thoroughly discuss our results, particularly those deemed significant, while approaching them from a causal perspective and comparing them with existing literature. However, due to the extensive range of results, including findings such as the one regarding solvents (OR 0.23, CI 0.01 to 1.05), which, despite having narrow estimates and intervals close to significance, were not initially deemed worthy of discussion based on our prior knowledge. Citing (Keil & Edwards, 2018, 437–38) That is, causal inference is an exercise in prediction…More generally, the field of causal inference has given rise to a particular type of prediction as the object of inference itself: potential outcomes. (Keil & Edwards, 2018, 437–38)

Minor comments

Page 5, line 130 – 141: The description of the imputation can be substantially condensed.

Thanks for the comment. I shortened it in the manuscript.

---

## [Decision Letter · Decision Letter 1]

18 Jun 2024

PONE-D-23-32546R1Maternal occupational exposures during early stages of pregnancy and adverse reproductive outcomes in the NINFEA birth-cohortPLOS ONE

Dear Dr. d'Errico,

Thank you for submitting your manuscript to PLOS ONE. After careful consideration, we feel that it has merit but does not fully meet PLOS ONE’s publication criteria as it currently stands. Therefore, we invite you to submit a revised version of the manuscript that addresses the points raised during the review process.

The reviewers has identified a few issues that would benefit from clarification.

We look forward to receiving your revised manuscript.

Kind regards,

Emily W. Harville

Academic Editor

PLOS ONE

Journal Requirements:

Reviewers' comments:

Reviewer's Responses to Questions

**Comments to the Author**

1. If the authors have adequately addressed your comments raised in a previous round of review and you feel that this manuscript is now acceptable for publication, you may indicate that here to bypass the “Comments to the Author” section, enter your conflict of interest statement in the “Confidential to Editor” section, and submit your "Accept" recommendation.

Reviewer #3: (No Response)

Reviewer #4: All comments have been addressed

2. Is the manuscript technically sound, and do the data support the conclusions?

Reviewer #3: Partly

Reviewer #4: Yes

3. Has the statistical analysis been performed appropriately and rigorously? 

Reviewer #3: Yes

Reviewer #4: Yes

4. Have the authors made all data underlying the findings in their manuscript fully available?

Reviewer #3: Yes

Reviewer #4: Yes

5. Is the manuscript presented in an intelligible fashion and written in standard English?

Reviewer #3: Yes

Reviewer #4: Yes

6. Review Comments to the Author

Reviewer #3: The study “Maternal occupational exposures during early stages of pregnancy and adverse reproductive outcomes in the NINFEA birth cohort” by d’Errico et al. investigates several occupational exposures assessed via self-report and two job-exposure matrices in relation to birthweight, preterm birth, and cesarian delivery. The authors address the potential impact of co-exposures by employing exploratory factor analysis. In the revisions, the authors were responsive to reviewer feedback and made several improvements to the manuscript. However, to be publication ready, I have suggestions for additional details and clarifications (outlined in the comments below). In particular, the authors should provide the sample sizes for exposed women who experienced preterm birth or cesarean section.

Major comments:

Table 4—Please provide the analytical sample size in the title. For each exposure, please provide the number of exposed women who experienced each outcome (preterm birth and cesarian delivery). Some of the associations are very imprecise and potentially unstable if the number of exposed women who experienced the outcome is small (e.g., < 5). This information is necessary for the reader to evaluate the associations. I agree with not relying on p-values and statistical significance testing to interpret results, however, there appears to be a potential issue with instability of some of the estimates.

There are many factors that determine whether a woman has a vaginal delivery vs. a cesarian section. What is the rationale for how these occupational exposures may impact risk of cesarian section?

Can the authors clarify how the analysis of EDC exposure is distinct from the cohort-specific associations reported by Birks et al.? A longer enrollment period? Why not include individual EDCs in the factor analysis?

Minor comments:

The authors might consider “birth outcomes” in the title rather than “reproductive outcomes” for consistency with the manuscript text.

Abstract/methods and Methods/confounders—The authors state they restricted to primiparous women, but should this be nulliparous?

Methods/Questionnaire—While the questionnaire items are provided in the supplementary material, it would be helpful to provide a few details and/or examples of the working load perception items, reduction of working hours, and physical exposures. Also, I do not see dust exposure in the questionnaire (Table S2).

Methods/Statistical analysis—estimating associations for each exposure separately does not constitute an ExWAS approach.

Results/study population—consider “without paid employment” as more inclusive of women who work inside the home, as “unemployed” may imply to some readers as seeking employment.

Results/JEM—I presume that the sensitivity analysis presented in Table S4 was intended to provide insight into the influence of exposure misclassification in the observed estimates. Thus, it is not clear why the authors chose a lower threshold for assigning exposure instead of higher threshold for classifying exposure which we would expect to reduce misclassification among the exposed group. Please clarify the rationale for the lower threshold and how this informs the interpretation of results.

Results/exploratory factor analysis—It seems from the authors’ presentation of the EFA results that the analysis discriminated between certain occupational groups. This makes sense, but it is less clear how these groupings and the associations with birth outcomes provides insight into the potential role of the exposures and co-exposure pattens in the analyzed outcomes (besides being indicative of occupational group). Can the authors comment more on this?

Table 4—Describing the factors in Table 4 in terms of the exposure patterns (rather than “F1”, “F2”, etc.) would be helpful for interpretation.

Reviewer #4: Previous comments appear to have been adequately addressed by revisions or providing rationale for choices. The manuscript analysis is adequate and does not require further revision.

7. PLOS authors have the option to publish the peer review history of their article (what does this mean?). If published, this will include your full peer review and any attached files.

Reviewer #3: No

Reviewer #4: **Yes: **Heather Young

---

## [Author Response · Author response to Decision Letter 1]

15 Oct 2024

Reviewer #3: 

The study “Maternal occupational exposures during early stages of pregnancy and adverse reproductive outcomes in the NINFEA birth cohort” by d’Errico et al. investigates several occupational exposures assessed via self-report and two job-exposure matrices in relation to birthweight, preterm birth, and cesarian delivery. The authors address the potential impact of co-exposures by employing exploratory factor analysis. In the revisions, the authors were responsive to reviewer feedback and made several improvements to the manuscript. However, to be publication ready, I have suggestions for additional details and clarifications (outlined in the comments below). In particular, the authors should provide the sample sizes for exposed women who experienced preterm birth or cesarean section.

Major comments:

1. Table 4—Please provide the analytical sample size in the title. For each exposure, please provide the number of exposed women who experienced each outcome (preterm birth and cesarian delivery). Some of the associations are very imprecise and potentially unstable if the number of exposed women who experienced the outcome is small (e.g., < 5). This information is necessary for the reader to evaluate the associations. I agree with not relying on p-values and statistical significance testing to interpret results, however, there appears to be a potential issue with instability of some of the estimates.

• Thanks for the suggestion. The frequencies of exposure by outcome have been included in Table 4 and the sample size has been added in the title.

2. There are many factors that determine whether a woman has a vaginal delivery vs. a cesarian section. What is the rationale for how these occupational exposures may impact risk of cesarian section?

• There are several medical reasons why a fetus may not be delivered vaginally. In our analysis, we focused exclusively on nulliparous women to exclude those who might have previously undergone a C-section, as prior C-sections increase the likelihood of subsequent C-sections. We included C-sections in our analysis to capture other potential pregnancy complications beyond preterm birth and birthweight.

For instance, fetal distress (McDonald et al., 1988; Liu et al., 2019), placental complications (such as abruption or previa) (Snijder et al., 2012; Raikkonen et al., 2014), and pre-eclampsia (Haelterman et al., 2007; Bonzini et al., 2007) are pregnancy complications associated with certain occupations and occupational exposures. For instance, Marcoux et al. investigated primiparous women in Quebec, Canada, and found that those exposed to high job strain (“high demand, low control”) had a significantly increased risk of developing pre-eclampsia and gestational hypertension. Both conditions contribute to a higher likelihood of C-section. Similarly, Hung et al. examined the impact of prepartum work practices on delivery type. Their study indicated that occupations involving prolonged standing, such as sales positions, and those requiring higher levels of physical effort, are associated with an elevated risk of cesarean delivery. 

This has been clarified in the manuscript (Methods - Birth outcomes section) and references have been added.

3. Can the authors clarify how the analysis of EDC exposure is distinct from the cohort-specific associations reported by Birks et al.? A longer enrollment period? Why not include individual EDCs in the factor analysis?

• Birks et al. found an association between endocrine-disrupting chemicals (EDCs) and low birthweight, with an odds ratio (OR) of 2.32 (95% CI 0.88 to 6.10) in a study of 1969 participants, of whom 155 were exposed. Similarly, our study observed a decrease in birthweight of -78.8 grams (95% CI -164.2 to 6.6) of whom 151 were exposed.

Our study included a larger sample size, with 3938 participants. Since we focused on nulliparous women, our study population differs from that of Birks et al., though there may be some partial overlap.

We did not analyze individual EDCs because were interested in assessing their potential associations with adverse birth outcomes as a group. Furthermore, for many individual chemicals within the EDC JEM, such as pesticides, polychlorinated organic compounds, phthalates, and BPA, most exposures did not have 20 exposed women, the thresholds we established for inclusion on the analyses.

Minor comments:

4. The authors might consider “birth outcomes” in the title rather than “reproductive outcomes” for consistency with the manuscript text.

• We have harmonized it in the manuscript. Thanks for the suggestion.

5. Abstract/methods and Methods/confounders—The authors state they restricted to primiparous women, but should this be nulliparous (ok nulli-primigravid)?

• Thanks for the remark, it is nulliparous. We have harmonized it in the manuscript.

6. Methods/Questionnaire—While the questionnaire items are provided in the supplementary material, it would be helpful to provide a few details and/or examples of the working load perception items, reduction of working hours, and physical exposures. Also, I do not see dust exposure in the questionnaire (Table S2).

• Thanks for your careful review. We have included an * for the explanation on why dust was added in the list even if it was not present in the original questionnaire.

We used the term 'perception' for workload because, in a questionnaire, it is difficult to quantify such factors from a quantitative or industrial hygiene perspective (e.g., using tools like the Borg Workload Scale or NASA-TLX). Assessing workload without relying on self-report is challenging, as much of human work involves mental activity that cannot be objectively measured (Webster et al., 2018). The questions are provided in the supplementary material, ensuring that readers have access to the same information available to us. 

7. Methods/Statistical analysis—estimating associations for each exposure separately does not constitute an ExWAS approach.

• We have clarified the approach used without denoting it as ExWAS, as suggested.

Results/study population—consider “without paid employment” as more inclusive of women who work inside the home, as “unemployed” may imply to some readers as seeking employment.

• We changed the terminology as you suggested.

8. Results/JEM—I presume that the sensitivity analysis presented in Table S4 was intended to provide insight into the influence of exposure misclassification in the observed estimates. Thus, it is not clear why the authors chose a lower threshold for assigning exposure instead of higher threshold for classifying exposure which we would expect to reduce misclassification among the exposed group. Please clarify the rationale for the lower threshold and how this informs the interpretation of results.

• We initially chose two thresholds that we considered reasonable (the 50th and 75th percentiles) for our analysis. We then opted to use the higher threshold to reduce misclassification among the exposed group, as you mention. However, in order not to deviate from the analysis plan devised a priori, we considered it useful to show the results for both thresholds considered. Additionally, we conducted a sensitivity analysis that included only jobs that could be classified with four digits ISCO to assess how imputation affected the estimates.

9. Results/exploratory factor analysis—It seems from the authors’ presentation of the EFA results that the analysis discriminated between certain occupational groups. This makes sense, but it is less clear how these groupings and the associations with birth outcomes provides insight into the potential role of the exposures and co-exposure pattens in the analyzed outcomes (besides being indicative of occupational group). Can the authors comment more on this?

• In an exposure assessment setting, defining women's exposure profiles is particularly useful when dealing with co-exposures. Apart from the first factor (F1), which can be considered a macro-group of healthcare workers (even though it is not a specific occupational group), the other factors can represent broader occupational groups with similar exposures and characteristics. The advantage of the EFA is that exposures can be quantified by the loading strength of each exposure variable (i.e., the correlation with the factor: positive or negative) within each factor, providing a good visualization for the readers as well. The individual scores derived for each participant provide insights into the participant's contribution to the factor, indicating ‘exposure levels’ within that group of exposures. Using EFA helps manage multicollinearity, which makes it difficult to account for the individual effect of each variable on the outcome using linear regressions, especially in a wide approach, and reduces the dimensionality of the numerous exposure variables while retaining most of the information. This method highlights the underlying structure of the data and captures complex relationships between exposure variables that might not be evident in the original feature space.

10. Table 4—Describing the factors in Table 4 in terms of the exposure patterns (rather than “F1”, “F2”, etc.) would be helpful for interpretation.

• As you suggested, we modified factors ‘titles’ for a better interpretation:

F1 = Healthcare chemicals and active work

F2 = Industrial chemicals

F3 = Physical-work and EDCs

F4 = Outdoor work

F5 = Workload and detergents 

F6 = Stressful work

---

## [Editor Report · Decision Letter 2]

18 Oct 2024

Maternal occupational exposures during early stages of pregnancy and adverse birth outcomes in the NINFEA birth-cohort

PONE-D-23-32546R2

Dear Dr. d'Errico,

We’re pleased to inform you that your manuscript has been judged scientifically suitable for publication and will be formally accepted for publication once it meets all outstanding technical requirements.

Kind regards,

Emily W. Harville

Academic Editor

PLOS ONE
---

## [Editor Report · Acceptance letter]

23 Oct 2024

PONE-D-23-32546R2 

PLOS ONE

Dear Dr. d'Errico, 

I'm pleased to inform you that your manuscript has been deemed suitable for publication in PLOS ONE. Congratulations! Your manuscript is now being handed over to our production team.

Kind regards, 

on behalf of

Dr. Emily W. Harville 

Academic Editor

PLOS ONE